# Towards Indicators for a Negative Emissions Climate Stabilisation Index: Problems and Prospects

**Mathias Fridahl [1],\* , Anders Hansson [1] and Simon Haikola [2]**

[1] The Centre for Climate Science and Policy Research and the Department of Thematic Studies, Unit of Environmental Change, Linköping University, SE-581 83 Linköping, Sweden; anders.n.hansson@liu.se

[2] Department of Thematic Studies, Unit of Technology and Social Change, Linköping University, SE-581 83 Linköping, Sweden; simon.haikola@liu.se

\* Correspondence: mathias.fridahl@liu.se; Tel.: +46-11-36-3184

**Abstract:** The incongruence between the United Nations objective to hold global warming well below 2 °C and the rate of global emission reductions has intensified interest in negative emissions. Previous research has explored several pros and cons of individual negative emissions technologies. Systematised approaches to comparing and prioritising among them are, however, largely lacking. In response to this gap in the literature, this article reviews the scientific literature on indicators for designing negative emissions climate stabilisation value indexes. An index typically provides summary measures of several components, often denoted indicators. Utilizing a narrative review methodology, the article derives five categories of indicators underpinned by overlapping and often mutually reinforcing environmental and socio-economic values. A list of 21 indicators are proposed to capture both positive and negative values associated with effectiveness, efficiency, scale, risk, and synergies. While discussing indicators capable of providing guidance on negative emissions is timely, given the emerging shift away from pure emission reduction targets towards net-zero targets, numerous complexities are involved in determining their relative values. The results herein serve to inform policy making on the prioritisation and incentivisation of negative emissions technologies capable of delivering on the new objectives, and the results highlight the many risks and uncertainties involved in such exercises. The article concludes that systematic research on the comparison of NETs is incomplete. An iterative, interdisciplinary research programme exploring such questions has the potential to be extremely rewarding.

**Keywords:** carbon dioxide removals; CDR; negative emissions technologies; NETs; value index; indicators

## 1. Introduction

Climate change continues to challenge conventional fossil-based development pathways. That governments across the world have begun to understand this challenge is reflected in the 2015 United Nations (UN) Paris Agreement to hold global warming well below 2 °C and to strive to limit it to 1.5 °C [1]. Hilaire et al. [2] note that the '[f]ast dwindling carbon budgets' (p. 190) for the temperature goal of the Paris Agreement first and foremost requires a rapid decarbonisation of the global energy system. Notwithstanding large decarbonisation efforts, meeting the temperature goal is also likely to require large-scale deployment of complementary solutions in the form of negative emissions technologies (NETs) [2,3]. The Intergovernmental Panel on Climate Change (IPCC) estimates that the carbon budget for the 1.5 °C is likely to be depleted by the end of the 2020s [4] (Figure 1).

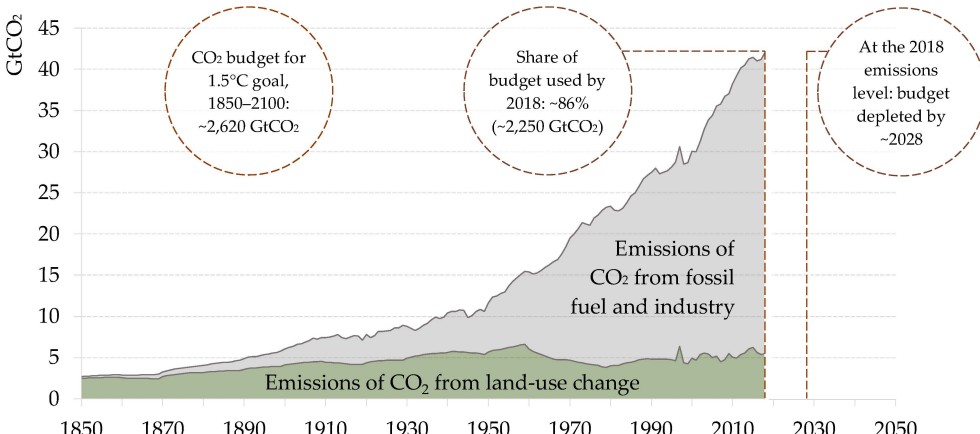

**Figure 1.** The total global carbon budget in the period 1850–2100, for limiting global warming to 1.5 °C by 2100, and the approximate share of the budget consumed by 2018. If emissions continue at the 2018 level, the global carbon budget will be depleted by about 2028. Any emissions thereafter would have to be compensated by carbon dioxide removals. Note that substantial uncertainties apply to the carbon budget estimate. Sources: IPCC [4], Gilfillan et al. [5], and Houghton and Nassikas [6].

The significance of negative emissions for limiting global warming well below 2 °C—preferably at 1.5 °C—is becoming even more obvious as adequate emissions reductions are conspicuous by their absence [3]. At the dawn of the 2020s, the significance of negative emissions is also manifest in an increasing amount of countries pledging net-zero emissions targets based partly on carbon dioxide removals. However, when moving from setting net-zero targets to discussing policy that enables their delivery, policymakers run into questions such as: What exactly constitutes a negative emission? Can negative emissions offset fossil emissions? How can various technological options to achieve negative emissions be weighed against each other?

Previous research has explored the pros and cons of specific NETs. Systematised approaches to comparing and prioritising among these technologies are, however, largely lacking. One approach to systematise comparisons is to develop indexes. An index provides summary measures of several components, often denoted as indicators. This article therefore—in an effort to start addressing the research gap on systematised comparisons of NETs—reviews the scientific literature in search of indicators to use for designing negative emissions climate stabilisation value indexes.

The list of indicators is intended to inform continued discussion on the topic of how to prioritise *between* NETs, not to be confused with prioritising future NETs *over* immediate conventional mitigation. Several commentators have warned of the hazard involved in deterring contemporary mitigation based on assuming the highly uncertain deployment of future NETs [7–9]. On the other hand, solving the climate crisis solely through conventional mitigation technologies also seems unrealistic. This necessitates continued technical and policy development and public deliberations [10,11]. The article can serve as a departure point for discussions of the importance of systematic assessments that are sensitive to site-specific determinants of value, such as informing policy making on the potential role as well as complexities of negative emissions in the global response to climate change.

Section 2 provides a brief background to NETs, presents the literature review method applied, and discusses how indicators can be used to construct indexes of value in relation to specified objectives. Section 3 describes the five categories of indicators—effectiveness, efficiency, potential, risk, and synergies—and Section 4 puts the complexities these indicators entail for constructing indexes in the context of needs for data and flexible governance structures. Section 5 concludes that even if the proposed indicators of value ought to be globally applicable, site-specific determinants of value undermine the potential to aggregate assessments to the global scale.

## 2. Background, Materials, and Methods

### 2.1. Negative Emissions Technologies

Most NETs are based on using photosynthesis as an engine for atmospheric carbon dioxide removals, for example, re- and afforestation, the increased use of harvested wood products, enhanced soil carbon, ocean fertilisation, and bioenergy with carbon capture and storage (BECCS). Some NETs are, however, based on chemical removals, for example, direct air carbon capture and storage (DACCS) and enhanced weathering [12]. Most photosynthesis-based NETs use terrestrial biomass, but the possibility to use marine feedstocks for BECCS or biochar is also discussed in the literature [13]. Ocean fertilisation is solely based on marine photosynthesis. The NETs also differ in their typical storage location and permanency. Some store carbon in stocks that require maintenance to remain unoxidised. Others are more maintenance-free with more or less perpetual storage [12,14], see Table 1. To, for example, execute the full technology chain of BECCS and DACCS means that most of the carbon dioxide injected into suitable geological storage formations is stored permanently or for long periods of time. However, such storage is being associated with a low but real risk of physical leakage and less carbon dioxide is stored if injected for purposes of enhanced oil recovery (EOR) [15–18].

The characteristics of various NETs can be explored in reference literature on the topic. For overviews of NETs, see, for example, IPCC [4], Fuss et al. [19], Minx et al. [12], and McLaren [20].

**Table 1.** Overview of characteristics of different negative emissions technologies.

| Negative Emissions Technology | Removal Process | Capture Medium | Storage (Time; Location) |
|---|---|---|---|
| Re-/afforestation | Photosynthesis | Land biomass | Temporary; increased carbon stock |
| Harvested wood products | Photosynthesis | Land biomass | Temporary; increased carbon stock |
| Soil carbon sequestration | Photosynthesis | Land or marine biomass | Temporary; increased carbon stock |
| Biochar | Photosynthesis | Land or marine biomass | Temporary; increased carbon stock |
| BECCS | Photosynthesis | Land or marine biomass | Long-lasting; geological |
| Ocean fertilisation | Photosynthesis | Marine biomass | Temporary to long-lasting; minerals (land) and sediments, or calcification (oceans) |
| Enhanced weathering | Chemical | Silicate/carbonate rocks | Temporary to long-lasting; minerals (land) and sediments, or calcification (oceans) |
| DACCS | Chemical | Amines or carbonation | Long-lasting; geological |

Sources: Modified from Minx [12] with additions from Samari et al. [14], Iordan et al. [21], and Fuss et al. [19].

### 2.2. Method: Narrative Review

To date, several thousand articles have been written that explicitly address negative emissions. Minx et al. [22] estimate that approximately 3000 articles on negative emissions were published in the period 1991–2016. Deriving indicators of value from a close reading of this literature obviously must be based on a sample. Following Small's [23] argument for the need to conduct selective literature reviews in fields comprising large numbers of publications, it is safe to say that while there are thousands of articles on negative emissions, they do not address thousands of different NETs. The sample in this article was selected to represent perspectives from a broad set of different sciences and literature that cover all the NETs discussed above. The aim is to attain comprehensiveness with respect to the range of merits and potential problems of applications of NETs and not the substantive specificities of different NETs applied in different locations. This approach aligns well with so-called *narrative reviews*, i.e., reviews guided by a broad aim to conduct an initial search, identify keywords to extend the search iteratively, and finally review abstracts and articles to identify narrative themes that broadly capture the literature through a selective, representative sample [24,25]. This approach was used to provide broad overviews of research fields and to avoid a more strongly reductionist approach that would zoom in on more specified issues. The merit of narrative reviews is to enable the assessment of large sets of literature, yet with the obvious drawback that such reviews cannot give justice to the multiple research frontiers on more specific issues within this broad span.

The review acknowledges that any discussion of value must start with more fundamental questions: valuable for whom, for what purpose, and when? Setting objectives and specifying parameters against which to assess value, and then selecting indicators to use as proxies for such values in effect constitutes an exercise in designing system boundaries. Location, timing, scope, and the level of ambition are key determinants of value—framing conditions—aspects that ought to be defined and specified before compiling various indicators into indexes and assigning relative weight to them in order to assess the value of the NETs.

The scope of the objective is fundamental to understand value. If the objective is to reduce emissions, it is not obvious that changes in albedo due to the deployment of NETs should be part of the assessment. While albedo is significant to the temperature response of NETs, it has nothing to do with the potential for actual carbon removal. Geographical location, similarly, can have dramatic effects on the relative value of negative emissions. For instance, deploying biochar to different soils may store similar amounts of carbon dioxide regardless of location. However, various co-benefits may be noticeably different. Biochar's contribution to increasing yields and reduced vulnerability to impacts of climate change is generally much higher if applied to more degraded, acid, compact, and drought-exposed soils. On the other hand, tropical regions are exposed to relatively high risks for biodiversity losses and reduced forest carbon stocks from ill-selected biochar feedstocks. Defining geographical boundaries for objectives is, therefore, key to understanding value.

The timing of objectives is also significant. With a longer timeframe, the value of negative emissions achieved through NETs with longer commissioning times and greater potential may increase, such as the negative emissions achieved with BECCS. In contrast, if the timeframe is short, more immediate measures become more pivotal. A single target year, as opposed to a compliance period or target trajectory, also has potential consequences, such as those related to the importance of the decay of stored carbon.

The level of ambition of objectives, finally, also provides a framework condition for understanding value. For example, aspiring to achieve large amounts of negative emissions may simply not be possible through otherwise highly valued NETs that have limited technical potential.

This article departs from global, long-term climate temperature stabilisation as the primary objective against which the value of negative emissions is assessed. This is not to say that NET abilities to bring values to other objectives are irrelevant. Avoiding silo mentalities in governance is key to preventing, or minimising, response measures designed to deal with one problem but that cause many other problems instead [26]. Potential trade-offs between using negative emissions to achieve climate stabilisation and sustainable development goals (SDGs), human rights, and ethics have been discussed in the literature and are highlighted in this article as an important category of potential indicators with major repercussions for value, yet the scope of which has not been fully explored due to the above delimitations.

Designing Value Indexes

Indexes are commonly applied to summarise and often compare entities in complex settings. Two common forms are stock market indexes and intelligence quotients, but indexes are applied to all sorts of issues such as assessing vulnerability to impacts of climate change [27], water stress levels [28], and the effects of diet on inflammation [29].

Indexes provide summary measures of several components, often denoted indicators. Such indicators are measures of quality related to an index. A vulnerability index, for example, may be comprised of indicators of exposure to climate variability, socio-economic determinants of adaptive capacity, and access to and dependency on resources to characterise sensitivity [27].

While an index often expresses something as an aggregate quantitative scoring derived through a relative weighting of individual indicators, designing a value index in the context of high and sometimes unquantifiable uncertainty is challenging [30,31]. This is the case when comparing the value of different forms of negative emissions, a young but nascent research field compared to research

on individual NETs. Lack of data and measurement standards constrain the potential to design robust indexes.

Two important conditions follow from this observation. First, the article identifies categories of indicators relevant to designing an index, but little practical guidance on how to perform the evaluation is given. Beyond pointing out key uncertainties, the article provides an analytic schema, the use of which will differ depending on context. While the index is set up for quantification, several of the indicators are inherently impossible to evaluate in any precise, quantitative terms, and numbers may only be derived from qualitative reasoning. Such numbers will vary depending on the purpose for which the index is used, and assigning numbers of this sort is ultimately a political decision. For example, the risk of carbon lock-in is inherently difficult to assess and will be rated differently depending on the scale and contextual particulars of the projects under consideration.

Second, an index of this sort should not be used to weigh negative emissions against immediate emissions reductions. This article aligns with the recommendation of several other researchers [8,32] and considers negative emissions separate from emissions reductions and, thus, avoids representing negative emissions and fossil emissions in a 1:1 relationship. It follows that the index should be applicable primarily in contexts where fossil fuel emissions are close to being phased out or where plans for such a phase-out already exist. This perspective concurs with McLaren et al. [8] who write that separate targets for emissions reductions and negative emissions should be maintained. It further follows that numbers are only possible to assign—tentatively—to NETs relative to each other, not to other technologies for managing climate risk, such as solar radiation management.

## 3. Results: Indicators of NET Values for Climate Stabilisation Objectives

The literature review resulted in a list of five categories of indicators—overarching narrative themes identified in the literature—focusing on effectiveness, efficiency, potential, risks, and synergies. Indicators pertaining to each of these categories are explored below.

### 3.1. Effectiveness

The category of indicators termed effectiveness emerged from a narrative theme in the literature targeting the ability of NETs to deliver the intended results, i.e., performance indicators [27]. Thus, the focus of this category of indicators is on the effectiveness of NETs in reducing temperature increase.

### 3.1.1. Global Cooling Potentials

Global warming potentials (GWPs) are consistently used to compare the amount of heat trapped by different greenhouse gases relative to the heat trapped by carbon dioxide. The cooling effectiveness of NETs can partially be expressed as negative GWPs or, in more pedagogical terms, as their global cooling potential (GCPs) [33,34].

As long as global emissions are net-positive, the GCP of a negative emission pulse could be considered inversely equal to the GWP of a positive emission pulse, simply because the cooling effect constitutes an avoided warming effect. Biogeophysically speaking, it would make sense to consider GWPs to equal GCPs until global emissions become net-negative. If and when global emissions turn net-negative, the GCP of a negative emission pulse would be less than the inverse magnitude of the GWP of a positive emission pulse of the same size [33]. In other words, even if the global net-negative emissions curve became an exact inverse copy of the previous net-positive emission curve, the resulting human-induced global cooling curve would not backtrack the previous warming curve. Earth system model simulations suggest that 'positive $CO_2$ emissions are more effective at warming than negative emissions are at subsequently cooling [the Earth]' (p. 1) [33]. Zickfeld et al. [33] have also found that the higher the atmospheric concentrations of greenhouse gases are before peak and decline, the less effective negative emissions are at cooling the Earth, mostly because the ocean surface heat and carbon dioxide sink capacity is reduced. Oceans function as a memory in the climate system; historical warming is carried into the future well past a hypothetical event in which global concentrations of

greenhouse gases are to be immediately returned to preindustrial levels [35,36]. However, it should be noted that the research on the global biogeophysical response to net-negative emissions is still very limited and that model results are in part inconclusive [37].

The consequences of findings by Zickfeld et al. [33] and others [36,37] have a significant impact on understanding carbon budgets for complying with the Paris Agreement's temperature goal. If the carbon budget for holding global warming well below 2 °C is temporarily exceeded before the budget is brought to balance at closure in 2100, this budget deficit must be repaid with interest. The amount of additional negative emission required to fully compensate for the warming induced by a carbon budget deficit is relative to the scale of the debt, i.e., higher temporary budget deficits generate higher interest rates. Delays between failed emission reductions and compensatory negative emissions thus put a one-to-one relationship out of play, which means that the earlier in time that negative emissions are achieved, the higher their value is for mitigating climate change.

The concept of GWPs has been used for policy purposes for decades. GWPs are applied to emission reduction commitments under the Kyoto Protocol to the UN Framework Convention on Climate Change. Kyoto Protocol commitments are expressed as percentage reductions compared to a reference year, and compliance is achieved through the net reduction of seven greenhouse gases. Each gas is weighted by its GWP and aggregated when measuring compliance. This procedure has been adopted despite knowledge of the arbitrariness of the GWPs' time horizons, i.e., applicability has been weighted against complete accuracy [38]. The same would inevitably be the case for any application of GCPs. GWPs are often expressed over 20, 100, and 500 years. For enhanced transparency and understanding, a range of time horizons would also be preferable for GCPs. Alternative metrics, such as global temperature change potentials, could be considered, but the benefit of GCPs would be their relative simplicity to calculate and their symmetrical comparability with the often used GWPs [39].

### 3.1.2. Removal Inertia, Storage Decay, and Storage Maintenance Requirements

Some NETs are characterised by gradual rather than immediate uptakes, for example, re- and afforestation. Since calculating GCPs depends on emission pulses, gradual removals of carbon dioxide following deployment reduces the immediacy of cooling effectiveness [40]. Because the timing of negative emissions is important to understand their GCP, removal inertia may be seen as negatively affecting the value of a NET.

In addition to removal inertia, some NETs are characterised by the gradual oxidisation of stored carbon [18,41]. Other NETs may suffer from the physical leakage of carbon dioxide [18,42]. Similar to the case of uptake functions, deriving globally uniform decay functions is close to impossible given the dependency on local circumstances of these functions. While it may be possible to derive a relatively accurate decay function for a specific form of biochar (pending feedstock and pyrolysis process parameters) applied to a specific soil (pending microorganisms, pH, moisture content, temperature, etc.), deploying another biochar to the same soil or the same biochar to another soil will most likely yield a different decay function [43–46].

From a broad systems perspective, the extent to which a photosynthesis-based NET achieves negative emissions depends on the sustainability of biomass production [47]. The IPCC accounting guidelines are, however, relatively clear. In the process of biomass growth, carbon is removed from the atmosphere, stored in biomass, and accounted for as a sink. When harvested, the carbon is accounted for as oxidised, either instantaneously or with some delay, depending on the intended biomass use (e.g., fuel or construction material). The overall growth or decline rates of biomass are the main determinants for whether the land use and forestry sector are carbon negative or positive. With this accounting logic, the carbon storage that comes about through the use of biomass outside of the land use and forestry sector automatically becomes a negative emission. As noted, however, systems performance depends on the capability of the land use and forestry sector to act as a continuous carbon sink. From this perspective, the decay function of stored carbon, rather than biomass regrowth, determines the stability of the negative emission.

For some NETs, such as soil carbon sequestration, continuous maintenance is required to preserve the stored carbon stock. Other NETs require less maintenance, such as biochar or carbon dioxide stored in adequate geological formations. The storage decay function, therefore, may be more or less dependent on the level of storage maintenance, which then can be used as an indicator of value.

Decay is partly dependent on sensitivity. For example, a diverse forest, with a mixture of deciduous and conifer trees, is normally less sensitive to extreme winds, fires, fungal diseases, and insect herbivores than a conifer mono culture [48]. Since pyrolysis results in a char that is not completely combusted, another example is the sudden oxidisation of biochar, which has been reported as prone to fire [49]. Sensitivity is less of a problem if the exposure to stressors is low. Factoring in exposure and adjusting land use and forestry practices to reduce sensitivity, may help increase the stability of stored carbon and reduce maintenance requirements.

### 3.1.3. Changed Albedo

Albedo ($\alpha$) denotes the capacity of a surface area to reflect incoming sunlight (expressed from 0 to 1). A high albedo denotes high reflection, a low albedo denotes more absorption and, thus, the generation of heat.

The albedo change ($\Delta\alpha$) of NETs has repercussions on local and global temperatures. This is of particular importance for area-intensive NETs, such as afforestation. Consider replacing a large light grass area ($\alpha \approx 0.25$) usually covered in snow during winter ($\alpha \approx 0.80$) with relatively dark spruce through afforestation [50]. However, $\Delta\alpha$ may be significant for less area intensive NETs too, such as applying biochar to soils even at more limited scales. Low-tech pyrolysis releases long-lived and strongly light absorbing soot, and biochar often darkens soils [43,44,51–53].

Hence, $\Delta\alpha$ can considerably affect the climatic response to the same amount of removed carbon dioxide, i.e., negative emission, depending on the choice of NET. To give one example, Meyer et al. [53] conclude that $\Delta\alpha$ may reduce the overall climate benefit of biochar by as much as 13–22% based on field laboratory tests of samples from a German biochar field combined with modelling. Smith [51] highlights that other NETs are associated with no $\Delta\alpha$, such as enhanced soil carbon sequestration through no-till agriculture and DACCS. According to Fuss et al. [19], the literature is also highly congruent on judging high latitude boreal re- and afforestation partly counterproductive due to reduced albedo.

Given the site-dependency of $\Delta\alpha$, attempts at deriving globally uniform estimates ought to be understood as crude approximations, complicated by the fact that incoming shortwave radiation differs by location, season, and type of above-ground biomass coverage. Surface albedo is also affected by factors such as soil humidity and cloud formation [43,53,54]. Despite these complexities, however, several attempts have been made to calculate the GWP of $\Delta\alpha$ [55].

### 3.1.4. Change in Direct and Indirect Fluxes of Greenhouse Gases

NETs are sometimes associated with an emissions penalty of greenhouse gases other than carbon dioxide, for example, a direct increase in emissions of nitrous oxide when CCS is deployed under certain conditions [56] or an increase in the release of methane from oceans following ocean iron fertilisation as a result of an increase in the digestion of phytoplankton [57]. There is also potential for direct emissions reductions of other greenhouse gases, for example, nitrous oxide from agriculture following the application of biochar [53].

In addition, indirect emissions may arise from the deployment of NETs. For example, if demand for negative emissions through harvested wood products, biochar or BECCS become drivers of unsustainable land use and forestry practices, then the carbon negativity of these NETs, over time, could be questioned [13,47,58,59]. While such indirect emissions would normally be accounted for as direct emissions from land use and forestry, connecting the dots between the causes of unsustainable land use and forestry and NETs is complicated with the current reporting structure, making it hard to track causal relationships.

Given current incentive structures, however, bioenergy demand is more likely to drive land use and forestry emissions than BECCS [60]. Adding a CCS component to bioenergy operations provides shared global collective climate benefits, yet with no or few possibilities to generate income to cover the specific private costs for the BECCS operator. In the event that bioenergy demand is a driver of unsustainability in the land use and forestry sector, BECCS has the potential to improve the total emissions performance.

While the above examples are related to indirect emissions from the production of feedstocks, indirect emissions may also result from the need to power CDR. DACCS and BECCS require high amounts of energy for capture, compression and cooling, transport, and storage. Enhanced weathering typically requires energy for crushing and transporting minerals. Some of these activities may give rise to indirect emissions, i.e., emissions that are accounted for elsewhere in the system but that are a direct consequence of activities associated with creating a negative emission. The net climate benefit depends on the broader energy system characteristics, such as if transportation is fossil-free or not. This is also true for substitution effects, or for the carbon leakage associated with increasing the production costs of deploying NETs. As Downie et al. [61] note in relation to biochar, but which holds true for all NETs: 'Accurate calculation of the net GHG [greenhouse gas] benefit of bioenergy production relies on the selection of the appropriate fossil reference system for the area in which the project is implemented' (p. 236). If assessing indirect emissions of NETs for purposes of including these functions as an indicator of value, the methodology must be selected with care.

### 3.2. Efficiency

While effectiveness focuses on possibilities to deliver desired outcomes, a key narrative theme in the literature is related to the efficiency by which such outcomes can be delivered. Efficiency indicators typically relate units of input to units of output [62].

### 3.2.1. Energy Efficiency

Energy conversion efficiency is a measure of the ratio between useful output energy and input energy. In the case of NETs, this is often expressed as the amount of stored carbon per unit of input energy [63,64]. All NETs require some form of input energy to power storage, although to different degrees and in different forms. BECCS or DACCS require the input of relatively large quantities of thermal, electrical, or mechanical energy to release carbon dioxide from a capture substance. In contrast, re- and afforestation, or enhanced soil carbon, may store carbon at very low levels of converted energy [63–65].

The energy efficiency of NETs can be used as an indicator of value, which in combination with factoring in the possibility of integrating NETs in symbiosis with existing processes (see Section 3.3.3), captures a useful measure of the NET energy penalty [66].

### 3.2.2. Resource Intensity (Land, Water, and Nutrients)

In a climate-constrained world, the demand for biomass is likely to increase [67,68]. As biomass production often competes with alternative land use practices [13,69], land quality as well as land area requirements per unit of stored carbon can be an important indicator of value. The ability of a NET to utilise land efficiently and to utilise marginal and otherwise unproductive land is valuable. Some NETs are less land intensive, such as ocean fertilisation and enhanced weathering. Aquatic BECCS, with algae-based feedstocks, or BECCS using waste feedstocks, typically have lower land use intensity relative to equivalents based on energy crops. The incremental energy demand driven by BECCS or DACCS can also be more or less land intensive depending on the primary energy source utilised [66].

Although value will differ with national or regional circumstances, from a global perspective, the availability of land is likely to become a more pronounced issue as demand for bioenergy and food is likely to increase with time.

The amount of water used per metric tonne of stored carbon may also indicate value. This is especially true in water-scarce areas but may become more significant in areas currently suffering low water stress, as climate change impacts weather patterns. As noted by Smith et al. [70]: 'Estimates of water required per t Ceq removed by DAC [direct air capture] and EW [enhanced weathering] are an order of magnitude or more lower than for BECCS' (p. 44). This points to the very different amounts of water required for different NETs. The largest part of water consumed for BECCS is related to feedstock. Thus, in processes already using biomass, it may be unfair to associate this water requirement with the carbon stored and instead focus on the relatively smaller part of water required for the CCS component of the BECCS. Aquatic biomass used for BECCS may also reduce the need for high quality fresh water [71]. However, the water requirement for additional cooling associated with energy produced to power CCS can be substantial [72]. DACCS may require small but substantial water inputs too [66]. While afforestation is evaluated as water intensive, it should be noted that forests also provide ecosystem services with positive values in relation to the water cycle. Biochar, as well as enhanced soil carbon, may add value in this context as the water retention capacity of soils improves with increasing amounts of organic matter and with the cell structure of biochar [73–75].

Phosphorus is already a scarce nutrient globally, and it is vital to maintaining productivity in agriculture [76]. Nitrogen and potassium deficiencies cause problems too, although the availability of these nutrients is greater than that of phosphorous. Nutrients may be a limiting factor for carbon storage potential when biomass grows. Adding nutrients may increase the carbon removal pace and scale, which is the underpinning logic of ocean fertilisation [57]. However, if biomass is harvested and transported away from its production site, nutrients will also be removed. Different forms of biomass contain different amounts of nutrients. Smith et al. [70] conclude that 'Bioenergy feedstocks with low nutrient concentrations, such as residue, forest and lignocellulosic biomass, should hence be favoured over feedstocks with higher nutrient concentrations' (p. 47). Similarly, land-based enhanced weathering may even add essential minerals to soils, which could also be seen as value added [77].

### 3.2.3. Cost

For any limited amount of financing, it makes sense to try to maximise the amount of negative emissions. Achieving cost-efficient response measures is key in the literature on negative emissions [19,49,78]. Appropriate weighing of a cost indicator with other proposed indicators is vital. Benefits of some NETs may be substantial, including high cooling effectiveness and high potential (see indicators proposed in Sections 3.1 and 3.3), and some NETs may have co-benefits with other policy objectives (Section 3.5). Some costs are also possible to cover through revenues once various benefits have been monetised in markets, while no markets currently exist for covering other costs, making them dependent on subsidies or similar policy interventions (Section 3.3.5).

The time dimension, which is linked to the stability of stored carbon, also becomes a significant factor when considering cost-efficiency. Some NETs require indefinite operation to maintain carbon storage. Agricultural practices used to enhance soil carbon [79] are an example of this. In cases where such practices make production uncompetitive, storage will depend on the potential to raise alternative financing, such as from public sources. While the capital costs involved may be low, the operational costs of such carbon storage accumulate over time. Other NETs may suffer from high initial capital costs yet may be cost efficient in a longer perspective [19].

Analyses of cost-efficiency are, thus, highly dependent on a priori assumptions and spatio-temporal delimitations. When used as an indicator of value, the variables used for analysis should be made transparent. Nevertheless, the cost of a NET becomes one important indicator of value among others, expressed as cost per unit of stored carbon.

### 3.3. Scale

While efficiency is key to delivering maximum effect with minimum input, even the most efficient NETs are limited by their potential to reach scale. This is discussed at length in the literature, thematised herein as a category of indicators of value related to scale defined by several limiting factors.

#### 3.3.1. Capture and Storage Potentials

The rapidly diminishing carbon budgets for holding global warming well below 2 °C indicate that immediate emission reductions and the deployment of near-term negative emissions must be paralleled by immediate capacity building for the long-term, large-scale deployment of NETs. This makes the scale of the technical potential of a NET relevant as an indicator of value, with pronounced applicability if value is measured against long-term objectives (such as the Paris Agreement's temperature goal) rather than mid-term objectives. A relatively immature NET with a longer implementation time but at potentially larger scale may, in a trade-off situation, be preferable to small-scale options. As noted by Pratt and Moran [49], cost-efficient NETs may be limited by their abatement potential and, therefore, may not be sufficient to reach specific targets.

The scale of potentials also relates to storage. Some NETs store carbon in structures that over time tend to become saturated, for example, carbon stored in forests, agricultural soils, construction materials, and even geological formations [69,70]. The significance of storage capacity is likely to increase with the level of ambition of climate objectives, depending on national circumstances (e.g., availability of land for re- and afforestation, suitable locations for biochar deployment, potential for soil carbon in agriculture, access to suitable geological formations for carbon dioxide storage). As noted by Smith [51], the sink saturation of soil carbon sequestration (SCS) 'occurs after 10–100 years, depending on the SCS option, soil type and climate zone' (p. 1323).

#### 3.3.2. Non-Rivalrous, Complementary Negative Emissions

The scale of the technical potential of one NET may be limited by rivalling NETs, such as competition for biomass resources for re-/afforestation, soil carbon sequestration, for use in BECCS facilities, or for biochar production [19]. The ability of a NET to coexist with other NETs may constitute an additional indicator of value. Fuss et al. [19] have proposed that enhanced weathering may be 'particularly valuable' (p. 9) because it is a non-rival to BECCS. This is an argument that should also be applicable to enhanced weathering's compatibility with other types of photosynthesis-based NETs. Levihn et al. [80] also highlight the potential for BECCS and biochar production to coexist, arguing that they 'do not cannibalize, but rather complement each other' (p. 1388) through their ability to utilise biomass for pyrolysis that is unsuitable for large-scale combustion.

NETs may not only be non-rivalling but also complementary, i.e., mutually reinforcing. Biochar and afforestation are examples of this; biochar reduces the sensitivity of forests to the impacts of drought, and forests reduce the sensitivity of biochar to being flushed out during flooding.

#### 3.3.3. Technical Integration

NETs also require integration into existing sociotechnical systems. The transition literature repeatedly reports on sociotechnical regime resistance to change [81,82]. Fajardy et al. [83] point out that such integration constitutes a 'challenge' (p. 3) to both BECCS and DACCS. Regime resistance and challenges with integration of new technical solutions, however, ought to exist for most NETs.

One option for energy intensive NETs is to make use of excess mechanical or thermal energy through smart process integration. Levihn et al. [80] demonstrate the importance of technical preconditions for BECCS deployment through integration in existing heat and electricity production systems, including the potential to use excess energy to capture carbon dioxide. Energy recovered from this process can be utilised in other ways, such as for district heating (a conclusion supported by other research [65]).

It is important to note that high potential for technical integration may not always connote positive value. This complicates the use of this indicator to assess value. The ability to integrate NETs in existing technical systems may further lock in dependency on technology that would otherwise be phased out due to negative impacts on sustainable development. BECCS and DACCS may strengthen the case for fossil CCS through the potential benefits of scale associated with shared infrastructure [84]. This position, however, is also criticised for its potential to prolong the fossil era [60,85]. BECCS and DACCS may also be associated with fossil extraction through EOR [15,86,87]. The latter could be captured by other proposed indicators, such as those related to indirect emissions, carbon leakage or displacement. The former type of potential effects of NETs deployment, for example, prolongation of the fossil era, are harder to assess since causality is often both relatively complicated to showcase and seldom one-directional.

The compatibility of NETs with existing technical regimes, thus, may also constitute an indicator of value as compatibility allows for a comparison of the relative ease by which NETs may be integrated into existing infrastructure. This indicator is closely related to the indicator on competition and is complementary with other NETs (see Section 3.3.2). Sociotechnical systems that already include a noticeable share of NETs overlap substantially. If both indicators are used in an index, attention should be paid to these potential overlaps and their effect on the overall scoring in the index.

### 3.3.4. Juridical Compatibility

While the potential for smooth technical integration increases the chances of a NET being implemented and scaled up, sociotechnical regimes also tend to resist change in many other ways, including the lock-in effects caused by existing regulation [81,88,89].

The fact that the regulatory environment for NETs is multilevel is likely to add to the slow pace by which regulation is changed to accommodate NET deployment [60,90]. The Parties to the International Maritime Organisation's London Protocol have managed to agree on how to circumvent the organisation's own export ban on carbon dioxide intended for sub-seabed storage [91]. While the Parties agreed to allow such export as early as 2009, the amendment has not entered into force due to the slow pace of ratifications [92]. In October 2019, the Parties decided that, pending the fulfilment of some procedural criteria, the amendment could be provisionally applied [93]. This development has reduced legal barriers where DACCS and BECCS are dependent on the potential to export captured carbon dioxide.

The situation for ocean fertilisation, on the other hand, has worsened. The scope for ocean fertilisation is limited by a moratorium adopted by the Parties to the Convention on Biological Diversity (CBD) in 2008. Although the moratorium is not legally binding [94], the moratorium requires adequate scientific basis on which to justify ocean fertilisation before deployment. This moratorium was upheld by the CBD in 2015 and supported by the London Protocol that prohibits deployment other than for permitted scientific purposes.

In many cases, NETs involve activities spanning several jurisdictions. BECCS may not only involve the export of carbon dioxide, the process may also involve the import of foreign biomass that may in turn be regulated both in the country of production and, in the case of the EU, through supranational regulation. This makes the value chain of some NETs more complicated than those with a more confined geographical scope.

While UN regulation significantly impacts the scope for NET deployment, national, supranational, and regional laws also influence the potential for deployment. The Convention on the Protection of the Marine Environment of the Baltic Sea Area (governed by the Baltic Marine Environment Protection Commission) prohibits geological storage of carbon dioxide under the Baltic Sea, and the EU's CCS directive [95] similarly prohibits storage in complexes that stretch beyond its member states' territories. This most likely applies to storage in the southern Baltic Sea, which has storage complexes that stretch into Russian territory. Given the investment risks associated with the incompatibility of NETs

with regulatory and administrative regimes, the level of juridical compatibility can be seen as a value indicator.

### 3.3.5. Market Compatibility

While cost efficiency provides measures for the relative cost of producing NETs, market integration has more to do with the possibility of receiving revenues to cover these costs. The extent to which NETs provide business opportunities may be an indicator of value as a business opportunity lessens the need for creating enabling environments through policy interventions. Biochar is already sold on markets throughout the world as a soil amendment to improve crop yields, reduce the need for fertilisers, and improve drought resilience [43,96]. When the sociotechnical system allows, excess heat from pyrolysis may also be recovered as district heating, thus improving the business case for biochar [97]. The added value of atmospheric carbon removals may improve the business case for a NET if such can generate value that costumers are willing to pay for, potentially including carbon credits [20,85,98–100].

### 3.3.6. Acceptance

Arguments in favour of a specific NET are undermined if the technologies stand little chance of being implemented due to low political, industrial, or social acceptance. Low acceptance for specific NETs is seen as a real barrier to deployment in the literature [49,60,85,100–102]. The level of acceptance, thus, may be included in an index as an indicator of value.

### *3.4. Risk*

A fourth category of indicators of value derived from a narrative theme in the literature focuses on risks. Effective and efficient delivery at scale my indeed be undermined by risk of failure, but the literature also underscores that NETs also have the potential to reduce system risks.

### 3.4.1. Diagnostic and Prognostic Uncertainty

While climate change in general requires decisions to be taken under deep (unquantifiable) uncertainty [103], NETs also suffer from more constrained and quantifiable uncertainty related to the specifics of the individual technologies. It is incredibly hard to predict how much carbon that is actually sequestered by growing forests in different locations around the world. This is not only due to measurement (diagnostic) uncertainty but also to future events such as potential drought and wildfires, storm-felling, pests, and illegal logging [68]. Jonas et al. [104] refer to such events as giving rise to prognostic uncertainty, i.e., the inability to accurately estimate true emission removals following the deployment of NETs.

The often adopted precautionary principle in international environmental law, combined with a lack of knowledge of various NETs' precise cooling effectiveness, uptake rates, saturation, and decay functions, etc., speaks in favour of creating uncertainty margins for assumed cases 'when true GHG emissions are smaller than inventoried estimates' (p. 10) [104]. The need for such margins is inversely dependent on the certainty that a NET can provide negative emissions. A high level of diagnostic and prognostic certainty, thus, could be considered as an indicator of value; greater certainty reduces the need for precautionary measures.

### 3.4.2. Investment Risks

Investing in NET capital involves risks. Some risks are related to the possibility to reap returns on capital and operational expenditure on markets partially approximated by the indicator of market compatibility (see Section 3.3.5).

However, there are risks involved in failed delivery due to technology maturity, scale, and complexity, including complexities involved in technical integration (Section 3.3.3). Less mature and more complex technology with high specific costs typically increases investment risks.

Market risks related to changes in interest rates or currencies tend to be more pronounced for long-term projects for which the economic development scenarios are more uncertain. Therefore, market risks often disfavour technology for which the time between an investment decision and full operation is relatively long. This is often the case for less mature and more complex technology and, thus, reinforces risks associated with technology.

Estimating investment risks may serve as a proxy for risk of failed delivery. If a certain amount of negative emissions is requested by a state to comply with an emissions target, investment risks underscore a need to compensate for potential failed delivery by planning for higher amounts of negative emissions than otherwise required to achieve the stated objective(s) or by diversifying investments in NETs, thereby spreading risks. As indicated by this discussion, there is value in NETs with low investment risks as well as in risk hedging.

### 3.4.3. Climate System Tipping Points

In addition to the potential of early negative emissions to have higher GCPs than late negative emissions, Geden and Löschel [9] also describe how early negative emissions bring value by contributing to an avoided temperature overshoot. Since little is known of the thresholds of climate system tipping points, i.e., the temperature at which an irreversible major change of the climate system is induced, lowering the global temperature peak can be very valuable.

### 3.5. Synergies

Finally, the climate stabilisation objective does not exist in a vacuum. Synergies with other objectives—both positive and negative—are reported to be significant. Some have been addressed only scantly, such as the impact of NETs on human and indigenous rights and local communities [105]. Other synergies have been discussed at more length, such as synergies with SDGs. As this article focuses on deriving indicators for designing negative emissions climate stabilisation value indexes, the mitigation dimension of SDG13 (climate action) has been covered at length in Section 3.1 to Section 3.4. The proposed indicators also cover aspects of SDG12 that deal with sustainable consumption and production patterns, such as SDG12.2 on the efficient use of natural resources. A focus on climate stabilisation, however, does not preclude using other sustainability indicators drawn from the SDG or human rights frameworks [31,106,107]. The literature on NETs report on SDG synergies in areas such as poverty eradication, food security, health, equality within and among countries, and gender [4,59,105,108]. Although NET synergies have been insufficiently studied, a few studies have focused on a generally applicable level. Examples include negative synergies between health and amine-based carbon capture solvents [109,110], the release of carbon monoxide and particles from low-tech biochar production [61], and risks of the abrupt release of large amounts of carbon dioxide from transport or storage associated with (BE)CCS and DACCS [111]. Photosynthesis-based NET synergies with adaptation, land degradation, and food security have been explored in slightly more detail [59,112], especially in relation to BECCS and re- and afforestation [4,19,59,112].

The literature conveys at least two general messages. First, it underscores the context dependency of NETs. The IPCC special report on land [59] points towards substantial negative synergies between large-scale BECCS deployment and adaptive capacity, desertification and land degradation, and food security. The report, however, also implements magnitude thresholds that show how small-scale deployment using best practices may turn negative synergies into positive ones. Therefore, while various indicators adopted from the SDG framework can be used to assess the value of NETs, the assessment ought to be made in view of national or local circumstances and the intended scale of deployment.

Second, whereas confidence in the magnitude of impacts is generally low, confidence in assessments of the direction of impacts is greater. Large-scale global deployment of BECCS (>11 GtCO2 yr$^{-1}$), biochar (>6 GtCO2 yr$^{-1}$), or afforestation (>9 GtCO2 yr$^{-1}$) will impact food security negatively due to land-use trade-offs [59].

## 4. Discussion: Designing a Negative Emissions Value Index

Indicators have been used in policy making for at least 100 years and are often cherished for their merits in dealing with complex and cross-cutting environmental issues [28]. A negative emissions climate stabilisation value index could facilitate understanding how different NETs rank in different locations. This would allow accounting for the local circumstances with which the NETs interplay.

It should, however, be noted that the selection of specific indicators to be included in an index, the overall number of indicators used and their relative weighing, requires careful consideration. These factors all influence the sensitivity of the index. Sullivan and Meigh [28] recommend that the 'determination of the value of weights to be applied in an index [ . . . ] should be achieved through participatory consultation and expert opinion' (p. 71), an approach that has the potential to improve the robustness of an index. This recommendation is echoed by others, such as Barnett, Lambert, and Fry [30], who argue that participatory or expert elicitation processes are essential for increasing the legitimacy of an index. The importance of deliberative processes is further reinforced by calls from Bellamy [10] to incentivise NETs responsibly, including by taking into account public values and interests.

The data associated with individual indicators also require sufficient resolution and standardisation. Without data for some indicators, the rationale behind the selection of indicators included in an index may be undermined, with the potential need to identify alternative proxies [113]. The importance of standardised data, whether quantitative or qualitative, is key to increase commensurability among NETs, and this importance is pronounced if the index is used to compare the value of the same set of NETs in different locations.

The forms of indicators proposed in this article, summarised in Table 2, have been tailored to be generic enough to encompass the many forms of NETs available to contribute to climate stabilisation, while specific enough to meaningfully account for local circumstances. One identified problem with the construction of environmental change indexes is that their often high level of abstraction risks undermining their usefulness [30]. This line of argument is reflected in the discussion about the problems of assessing NETs positive and negative synergies on a global scale (see Section 3.5). Such assessments depend on the scale and location of implementation. It is possible that one and the same NET causes severe problems and faces trade-offs in one location, while the same NET would show substantial co-benefits and complementarity in another. Global averages often hide the details that make the assessment socially relevant. Therefore, overexaggerating the spatial dimension to which an index is applied should be avoided. In the words of Barnett, Lambert, and Fry [30], this does not 'preclude the aggregation of the results of many such studies' (p. 115), at least if aggregation is done in ways that move beyond quantitative average scorings to show spread and diversity.

**Table 2.** Summary of prospective indicators for designing negative emissions climate stabilisation value indexes.

| Component | Indicator | Climate Stabilisation Value |
|---|---|---|
| Effectiveness | Global cooling potential | Positive value is inversely proportional to time from present until deployment. |
| | Removal inertia | Positive value is proportional to the speed of the carbon removal function. |
| | Storage decay | Positive value is inversely proportional to storage decay rates. |
| | Storage maintenance requirements | Positive value is inversely proportional to maintenance requirements, exposure, and sensitivity of the stored carbon. |
| | Albedo change | Value is proportional to albedo change caused by a NET. |
| | Direct emissions | Value is inversely proportional to the direct emission of greenhouse gases caused by the deployment of the NET. |
| | Indirect emissions | Value is inversely proportional to indirect emissions caused by the deployment of the NET. |
| Efficiency | Energy efficiency | Positive value is inversely proportional to the energy requirement per unit of stored carbon. |
| | Resource intensity | Positive value is inversely proportional to the: |
| | | land area and quality required per unit of stored carbon, |
| | | water required per unit of stored carbon, |
| | | Phosphorus, nitrogen, and potassium required per unit of stored carbon. |
| | Cost | Positive value is inversely proportional to the cost per unit of stored carbon. |
| Scale | Technical potential | Positive value is proportional to the potential deployment scale of carbon removal from the atmosphere. |
| | Storage capacity | Positive value is proportional to storage capacity. |
| | Non-rivalrous | Positive value is proportional to the ability to deploy a NET without competing with other NETs. |
| | Technical integration | Positive value is proportional to the ability to integrate a NET into existing technical systems. |
| | Juridical compatibility | Positive value is proportional to the compatibility of a NET with existing juridical and administrative systems. |
| | Market compatibility | Value is proportional to the difference between specific costs of a NET and its capacity to raise revenues on existing markets. |
| | Acceptance | Positive value is proportional to the level of acceptance of a NET. |
| Risk | Diagnostic and prognostic uncertainty | Positive value is proportional to the level of (biophysical) certainty of delivery of negative emissions. |
| | Investment risks, technical failure | Positive value is inversely proportional to the risk of technical failure. |
| | Tipping points | Value is dependent on the timing of negative emissions relative to the global temperature peak, with more value for pre-peak than post-peak negative emissions. |
| Synergies | Multiple relevant indicators available | Value is proportional to positive and negative synergies with other policy objectives. |

## 5. Conclusions

This article reviewed narrative themes in literature on negative emissions technologies (NETs) in search of indicators of relevance to the design of negative emissions climate stabilisation value indexes. While such indexes are both needed and timely, designing them is an intricate endeavour. The complexities involved in assessing the systemic effects of conventional mitigation were found to be pronounced for negative emissions, not least due to the additional uncertainties related to sink capacity, carbon uptake rates, and storage decay functions. Additional uncertainties are also associated with the relatively understudied dynamic climate system response to global net-removals of atmospheric carbon dioxide. In view of these complexities, progressing the research on systematic assessment frameworks of NETs is both daunting and much needed.

The article identified 21 prospective indicators organised into categories of effectiveness, efficiency, scale, risk, and synergies. These indicators capture both positive and negative values and are generic enough to be globally applicable. The article also found that the value of NETs is many times extremely site-specific. Even if the indicators are globally applicable, the results of applying them are likely to differ substantially by location. This fact greatly undermines the possibility to generate indexes that aggregate data at a global scale, especially if data are summarised as global averages. These averages hide the spread of value that is likely to occur in larger systematised comparative case-study research.

Some general lessons can, however, be drawn from the literature:

1.  Immediate reduction in emissions provides more certainty than reliance on future negative emissions. While it is clear that both a reduction in emissions and negative emissions are needed to reach the Paris Agreement's temperature objectives, the potential for future negative emissions should not motivate postponing conventional mitigation.
2.  The relatively lower global cooling potential of delayed negative emissions compared to near-term negative emissions, and the benefits of limiting temperature overshoot, indicate that the sooner negative emissions are deployed the better. While research and development of less mature NETs with high potential is a key concern, it is critical to also reap possibilities for the deployment of more mature NETs in the near term, even if their abatement potential is limited.
3.  The scale and location of the deployment of NETs often determine whether synergies with other objectives are positive or negative. Identifying and showcasing co-benefits additionally drive NET deployment on existing markets and contribute to building new businesses around values that can be sold on premium markets. Concurrently, limiting negative emissions due to harmful synergies may provide false assurances. Negative synergies triggered by the deployment of NETs must be weighed against the negative impacts of climate change.

The context dependency of the value of negative emissions also highlights the need for site-specific knowledge and data. The context dependency also calls for sufficiently adaptive policy responses with an organisational flexibility capable of accounting for changes in contextual circumstances. However, flexibility has to be balanced against the need to create stable enough planning horizons to reduce the risk of investing in NETs.

Continued research on the set of appropriate indicators and their relative significance is required to make negative emissions value indexes operative. Current research has much to say about NETs applied in different locations. Systematic research on how different NETs play out in the same location, or how the same set of NETs play out in different locations, is incomplete. An iterative, interdisciplinary research programme exploring such questions has the potential to be extremely rewarding. Such a programme could become the locus for the challenging task of developing practices for operationalising a negative emissions value index. This operationalisation should include finding a manageable balance between, on the one hand, accuracy and robustness and, on the other, transparency and applicability.

**Author Contributions:** M.F. is responsible for the research design and wrote the main part of the article. A.H. and S.H. contributed to all parts of the article through commenting and as co-authors. All authors have read and agreed to the published version of the manuscript.

**Funding:** This research was funded by the Swedish Environmental Protection Agency, grant number NV-05228-19 and the Swedish Energy Agency, grant number 46222-1. The APC was funded by the Swedish Energy Agency.

**Conflicts of Interest:** The authors declare no conflict of interest. The funders had no role in the design of the study; in the collection, analyses, or interpretation of data; in the writing of the manuscript; or in the decision to publish the results.

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
