# Peer review of "Towards Indicators for a Negative Emissions Climate Stabilisation Index: Problems and Prospects"

_climate, doi:10.3390/cli8060075_

Round 1

Reviewer 1 Report

The article titled “Towards Indicators for a Negative Emissions Climate Stabilisation Index: Problems and Prospects” is very interesting and actual. It is an article with high interest to readers.

It is well written and I just can recommend minor changes to finalize it.

Please find my comments as follow:

Line 30: remove space “2_ºC”. Revise the entire document.

Line 65: BECCS? Explain in the first time

Line 66: DACCS? Idem, revise the entire document.

Line 189: CO2 subscript.

Line 596: Table 2Fel! Hittar inte referenskälla. , have been tailored to be generic enough to encompass the many… correct.

Author Response

Dear reviewer, 

Thank you for your constructive comments. We have revised the manuscript accordingly by: removing spaces befor °C (at 6 instances), explained BECCS and DACCS, subscript the number in CO2, and revised Table 2.

Best, Mathias Fridahl and co-authors

Reviewer 2 Report

Reviewer Comments for Manuscript: Towards Indicators for a Negative Emissions Climate Stabilisation Index: Problems and Prospects

General Remarks:

The manuscript presents a review of indicators for designing negative emissions climate stabilization value indexes.

The manuscript proposed 21 indicators categorized into five different categories that captured both positive and negative values.

The manuscript is well structured; however, the abstract should be standalone enough to present the aim, the method used, results as well as an outlook. Here the literature review method employed is missing as well as an outlook.

The author(s) used numerous terminologies and poorly abbreviate them.

In general, the study should be a good read for the general public.

Therefore, I recommend the manuscript being accepted for publication after some minor review.

Few Specific Remarks:

Manuscript

Introduction

31 ..”all the” please consider deleting this to enhance understanding.

40 ..” systematized” adjust tense accordingly.

40-41 Please restructure the sentence to aid comprehension.

44 ..” discussions” Singular or Plural?

65 …” indexes” what does this stand for. On the first occurrence should be written in full (abbreviated) and afterward the abbreviation.

66 the same as above

87 Mind your tenses.

148 Really?

343 it is better you abbreviate this “DACCS” and “enhanced weathering” at first occurrence and use abbreviation subsequently.

596 – 597 Please consider changing this.

Reviewer 3 Report

Abstract - the number "21" should be presented as  text.

The abstract and introduction should underline the novelty of this study, making reference to other recent similar studies (specific studies).

Introduction: Here you should introduce some text about climate change, GHG, etc.

At least one figure should be introduced in this article, e.g. the actual global GHG trend or/and global temperature.

Few more recent references should be introduced, e.g.: 
- The 2 °C Global Temperature Target and the Evolution of the Long-Term Goal of Addressing Climate Change—From the United Nations Framework Convention on Climate Change to the Paris Agreement, https://www.sciencedirect.com/science/article/pii/S2095809917303077

- Negative emissions and international climate goals—learning from and about mitigation scenarios https://link.springer.com/article/10.1007/s10584-019-02516-4

- Negative emissions technologies: A complementary solution for climate change mitigation, 

https://www.sciencedirect.com/science/article/pii/S0048969719315177

The terminology "Air capture" must be introduced.

The text must be carefully checked, see Line 596, format of Table 2 (?)

Line 664, change the term "hope", change the idea of this paragraph.

Author Response

Dear reviewer, 

Thanks you for the constructive comments. We have revised the manuscript to highlight the research gap that we're addressing. In connection to this, we have also introduced relevant literature on this specific issue already in the introduction (as suggested). In the previous version of the manuscript, we adressed this mostly in the background section.

The problem of climate change has been explored,icated in the introduction, including a figure on global emissions and a discussion of the global carbon budget for the 1.5°C goal.

Thanks also for noting that we had not spelled out the acronym DACCS the first time we used it. This has now been done. We have also revised Line 596, as suggested (i.e. "Table 2").

All suggested references have also been read and included in the text.

The paragraph starting on line 664 has been revised, as suggested.

Best, Mathias Fridahl and co-authors